# TGF-β1 in plasma and cerebrospinal fluid can be used as a biological indicator of chronic pain in patients with osteoarthritis

Yen-Chin Liu[1,2], Hung-Tsung Hsiao[2], Jeffrey Chi-Fei Wang[2], Tzu-Cheng Wen[3], Shiou-Lan Chen [4,5,6,7] *

1 Department of Anesthesiology, Kaohsiung Medical University (KMU) Hospital, KMU, Kaohsiung, Taiwan,
2 Department of Anesthesiology, National Cheng Kung University Hospital (NCKU), College of Medicine, NCKU, Tainan, Taiwan, 3 School of Medicine, College of Medicine, NCKU, Tainan, Taiwan, 4 Graduate Institute of Medicine & M.Sc. Program in Tropical Medicine, College of Medicine, KMU, Kaohsiung, Taiwan, 5 Department of Medical Research, KMU Hospital, Kaohsiung, Taiwan, 6 Drug Development and Value Creation Research Center, KMU, Kaohsiung, Taiwan, 7 College of Professional Studies, National Pingtung University, Taiwan

* shioulan@kmu.edu.tw

## Abstract

### Introduction

Previous studies have demonstrated that cytokines, transforming growth factor (TGF-β1), and brain-derived neurotrophic factor (BDNF) can impact the intensity of pain in rodents. However, the roles of cytokines, TGF-β1 and BDNF in humans with chronic pain in osteoarthritis remains unclear, and no comparison between plasma and central cerebral spinal fluid (CSF) has been conducted.

### Methods

Patients with osteoarthritis who were scheduled to receive spinal anesthesia were enrolled. The intensity of pain was evaluated with a visual analogue scale (VAS). In addition, patients with genitourinary system (GU) diseases and without obvious pain (VAS 0–1) were included as a comparison (control) group. The levels of TGF-β1, BDNF, tumor necrosis factor-α (TNF-α), and interleukin (IL)-8 within the CSF and plasma were collected and evaluated before surgery.

### Results

The plasma and CSF TGF-β1 levels were significantly lower in the osteoarthritis patients with pain (VAS ≥ 3) than in the GU control patients. Downregulation of plasma BDNF was also found in osteoarthritis patients with pain. The Spearman correlation analysis showed that the VAS pain scores were significantly negatively correlated with the levels of TGF-β1 in the CSF of patients with osteoarthritis. However, there was no significant correlations between the pain scores and the levels of BDNF, TNF-α, and IL-8 in either the CSF or plasma.

**Data Availability Statement:** Since participants did not provide consent for their data for purposes other than those described in the original study, the datasets used in this study cannot be made public.

However, the de-identified versions of the datasets used for the current study may available on reasonable request upon approval by the by the University's Institutional Review Board for the Protection of Human Subjects at National Cheng Kung University (NCKU) Hospital, Taiwan. Human Research Ethics Office, NCKU, No. 138, Shengli Rd., North Dist., Tainan City, Taiwan, Telephone: +886 6 2353535 ext 4830, n964441@mail.hosp.ncku.edu.tw (please quote project number IRB#B-ER-104-070).

**Funding:** This study was supported in part by grant NCKUH-10509003 from National Cheng Kung University Hospital, Taiwan and 105-2314-B-006-007-, 106-2314-B-006-026-MY2 (to YCL), 105-2628-B-037 -003 -MY3 (to SLC), from the Ministry of Science and Technology, Taiwan.

**Competing interests:** NO authors have competing interests

## Conclusions

TGF-β1 but not BDNF, TNF-α, or IL-8 may be an important biological indicator in the CSF of osteoarthritis patients with chronic pain.

## 1. Introduction

Pain is an unpleasant stressful sensation that is caused by an acute injury or chronic disease [1]. Persistent chronic pain has been suggested to be a chronic disease state. For example, underlying diseases such as arthritis are usually associated with chronic pain for weeks or even months [2]. In patients with osteoarthritis (OA), chronic pain persists due to the progression of the disease and becomes a disabling symptom of OA [3]. Therefore, selection of pain control agents and finding the relevant biological indicators are important in the treatment of OA. Up to the present time, many studies have focused on treatments for the chronic pain caused by OA, in the hope that a pain-relieving strategy will improve the symptoms and the daily functioning in these patients [4–6]. In addition, an increasing number of studies are focusing on related biological or proteomic changes in pain conditions to determine either the pain indicators or pain control targets. Thus, in this study, we assumed that gaining an understanding of the molecular basis of pain perception may provide new insights into the treatment of chronic pain caused by OA.

Although the mechanisms of chronic pain in patients with OA is not well understood [7], the inflammatory process in the joint and its surrounding tissue is assumed to play a role in the developing of pain in patients with arthritis [8]. In patients with inflammatory arthritis, treatment with anti-rheumatic drugs is effective in reducing inflammatory pain symptoms [9]. Previous studies have found that inflammatory cytokines in the blood are associated with chronic low back pain [10]. Furthermore, serum cytokines, including tumor necrosis factor-α (TNF-α) and interleukin 8 (IL-8), have been found to be higher in patients with sciatica than in healthy controls [10]. In animal pain models, TNF-α [11, 12] and IL-8 [13, 14] have been shown to be significantly increased and to contribute to chronic neuropathic pain. Furthermore, transforming growth factor (TGF-β1), as a neuronal protective factor with an anti-inflammatory effect, has been demonstrated to inhibit the pain process in chronic constriction injury (CCI) models [15] and in neuropathic pain models [16]. These studies present strong evidence suggesting that cytokines and TGF-β1 may serve as potential pain indicators. However, the role of cytokines in chronic pain perception in the peripheral blood and central nervous system (CNS) of human subjects remains unclear. Evaluating the expression of cytokines and TGF-β1 in the blood and CNS may provide a better reference for the evaluation or treatment of chronic pain in clinical settings.

On the other hand, brain-derived neurotrophic factor (BDNF) can increase the intensity of pain in rodent inflammatory and neuropathic pain models [17, 18]. BDNF has been suggested to be the driving force for neuroplasticity in central sensitization in neuropathic pain models [17]. Conditional BDNF gene knockout mice have also demonstrated its involvement in chronic inflammatory pain [18]. These reports indicated that BDNF plays an important role in pain. However, the controversy surrounding the effects of inflammatory/anti-inflammatory cytokines and neurotrophic factor in pain remains. Thus, understanding the relationship between these molecules and pain perceptions are important, especially in the CNS and peripheral blood.

Therefore, in this study TGF-β1, TNF-α, IL-8 and BDNF were simultaneously collected from the cerebrospinal fluid (CSF) and plasma samples taken from patients with OA with (VAS score ≥ 3) or without (VAS score ≤ 2) significant clinical pain. The relationship between pain perceptions and molecular expression in CSF and plasma was evaluated and compared. We hypothesized that cytokines, TGF-β1 and the neurotrophic factors in CSF and plasma are correlated to the intensity of pain and that these molecules in subjects with chronic OA experiencing pain may serve as pain indicators of chronic pain in OA.

## 2. Materials and methods

### 2.1 Ethics approval and consent

The trial was registered prior to patient enrollment at clinicaltrials.gov (NCT03606915, Principal investigator: Yen-Chin Liu, Date of registration: 2015-05-22). This study was approved by the University's Institutional Review Board (IRB #B-ER-104-070) for the Protection of Human Subjects at National Cheng Kung University Hospital and Tainan Hospital of Ministry of Health and Welfare, Taiwan. The study procedures were performed according to the rules of the IRB and fully explained to the participants. After the study had been completely described to the participants, informed consent was obtained from all patients in the study. This study was performed in accordance with the standards of ethics outlined in the Declaration of Helsinki.

### 2.2 The design of the study

To evaluate the changes in the biological molecules in patients at different levels of pain, patients with or without chronic pain who had received spinal anesthesia for lower body surgery were enrolled in this study. The patients' pain intensity was evaluated based on the pain score prior to surgery. The levels of TNF-α, IL-8, TGF-β1, and BDNF both in CSF and plasma were also evaluated.

### 2.3 The selection of the patients

We collected patients with a pain history from 07/2015 to 02/2017 before they received spinal anesthesia for knee/hip OA surgery. Moreover, patients with genitourinary system (GU) diseases without obvious current pain at rest served as a comparison (control) group. None of the patients were pregnant; they had an American Society of Anesthesiologists physical status of 1 to 3 and were scheduled for lower body surgery. The inclusion criteria included 1) patients undergoing spinal anesthesia who were over 20 years of age, 2) patients who were able to understand the purpose of the study and had provided written informed consent. The exclusion criteria were 1) patients with contraindications for spinal anesthesia (refused spinal anesthesia, coagulopathy, or severe aortic stenosis) or refusal to participate in the study, 2) patients with pre-existing neurological diseases, 3) patients using immunomodulatory drugs, and 4) patients over 70 years old or with cancer. All patient medical records were reviewed.

### 2.4 Pain evaluation and analgesics used

We evaluated the intensity of pain before surgery. All patients were asked to evaluate their current pain intensity using the Visual Analogue Scale (VAS, ranged from 0–10, no moving or standing), based on their native language Mandarin or Taiwanese a day before the operation and biosample collection. The pain intensity was recorded by experienced nurses and was believed to be their current pain intensity as impacted by their disease. To confirm the situation of chronic pain, the duration of pain (days) was collected. Patients were asked to recall

how long the pain or discomfort caused by the disease has persisted. Complete blood count, liver and renal function, chest X-ray, and electrocardiogram were also routinely checked to exclude other medical conditions. The patients typically received their surgery and CFS sampling with the next few days. According to the patients' statements and their medical records, they were then grouped into 3 different groups: GU without obvious pain (control, VAS 0–1), OA with very mild pain (VAS $\leq$ 2), and OA with pain (VAS $\geq$ 3) based on their pain level a day before the surgery. The analgesic drugs used by the patients were also recorded based on their medical records within one month prior to sampling.

## 2.5 Blood and CSF collection

All patients were sampled in the operating room when they received spinal anesthesia for their scheduled surgery. All patients were collected in the daytime (8:30~16:30) and had no food and water intake for at least 8 hours. In the operating room, patients were in the lateral decubitus knee-chest position. After sterilizing and dressing the patient's lower back, we introduced a 25G spinal needle into the spinal canal and CSF (1 ml) was drained, collected, and then replaced with 0.5% bupivacaine (2–3 ml) as the spinal anesthesia. We also collected whole blood simultaneously before surgery began via venipuncture. Whole blood (10 ml) was drawn into a test tube that contained the anticoagulant ethylenediaminetetraacetic acid. The blood and CSF samples were ice bathed and then transported to a laboratory for a biological molecule analysis. The CSF and whole blood samples were then centrifuged (3000 × $g$ at 4°C for 10 min), and the CSF and plasma was isolated. The CSF and plasma were immediately stored at −80°C.

## 2.6 Measuring of plasma and CSF IL-8, TNF-α, TGF-β1 and BDNF, levels

The levels of IL-8, TNF-α, TGF-β1, and BDNF were quantified using enzyme-linked immunosorbent assays (ELISA) (HSTA00D for TNF-α, D8000C for IL-8, DB100B for TGF-β1 and DBD00 for free form BDNF, Quantikine Human Cytokine Kit; R&D Systems, Minneapolis, MN, USA) as before [19]. These kits were validated to measure human cytokines, BDNF in plasma [19] and supernatant of cell culture. In brief, the standard of molecules was established using recombinant human IL-8, TNF-α, BDNF and TGF-β1 (R&D) proteins diluted in buffer. In order to measure TGF-β1 and BDNF in the plasma, the plasma was diluted (1:10) with an assay diluent buffer before measurement. The relevant dilution factors were calculated back to the exact concentration of TGF-β1 and BDNF in the plasma. A total of 50 μl of either standard or diluted samples (plasma or CSF) was added to the plate. All laboratory assays were performed in duplicate. The ELISA plate was then incubated for 3 hours at room temperature on a rotary shaker and washed. After incubation with the conjugated buffer and substrate solution, the concentrations of the biological molecules were evaluated with an ELISA reader (SpectraMax-M2; Molecular Devices, Sunnyvale, CA, USA) and calculated by the concentration of standard.

## 2.7 Statistical analysis

Chi-square ($\chi^2$) tests, a one-way ANOVA with a Bonferroni post-hoc multiple comparison the demographic data. A multivariate analysis of covariance (MANCOVA) (covariates: age, gender, body mass index [BMI], hypertension, diabetes mellitus [DM] and the use of pain-relieving drugs) was used for the statistical evaluations. For the biological molecules are often non-normality, a non-parametric statistical test (Kruskal-Wallis test) was also chosen to analysis the data. To determine the correlation between the VAS pain scores and the plasma and CSF molecule, the Spearman correlations were calculated. SPSS 22.0 for Windows was used for the statistical computations. Significance was set at $p < 0.05$.

## 3. Results

We collected and analyzed 29 patients with end-stage OA (X-ray presented articular cartilage has worn away almost completely; bone on bone contact is occurring and passed another objective orthpedicst's check) when they received spinal anesthesia for total knee or hip replacement surgery. Our data demonstrated that some patients with chronic OA do not feel obvious pain when resting (VAS score 0–2, no moving or standing). Therefore, in order to understand the cause of different expressions of pain between patients, the OA patients were then divided into groups according to their VAS score as very mild pain (VAS score is $\leq 2$, n = 8) and pain (VAS score $\geq 3$, n = 21) groups. Moreover, 14 patients with GU diseases without obvious pain (VAS score 0–1 at rest) were included as a control group (Table 1).

There were no significant between-group differences in age, body weight, or body mass index (BMI) (Table 1). There were no statistical differences in the distribution of gender, hypertension, diabetes mellitus (DM), and pain-relieving drugs used for the groups (Table 1). The mean VAS pain scores for the GU control, OA with very mild pain and OA with pain groups were 0.3, 1.5 and 3.4 ($p < 0.0001$) respectively, which indicated the different degrees of pain intensity in these patients at rest. Moreover, to better understand the situation of chronic pain, the duration of pain (days) were recorded according to the recollection of the patients. The significant different duration of pain experience caused by the disease were found among GU control patients (mean duration of pain is 75.1 days), OA patients with very mild pain

**Table 1. Comparison of the general clinical parameters for the patients.**

| | GU patients | Chronic OA patients | | | $p$ |
|---|---|---|---|---|---|
| | VAS score 0–1 (control) | VAS score 0–2 (very mild pain) | VAS score $\geq$ 3 (pain) | | |
| N | 14 | 8 | 21 | | |
| Age (years, mean ± SD) | 57.3 ± 8.7 | 59.9 ± 6.6 | 62.2 ± 7.6 | F = 1.661[a] | 0.203 |
| Female sex, n (%) | 6 (42.9) | 7 (87.5) | 15 (71.4) | $\chi^2$ = 5.187[b] | 0.075 |
| Body weight (kg, mean ± SD) | 66.5 ± 6.9 | 73.1 ± 15.9 | 69.9 ± 13.1 | F = 0.793[a] | 0.459 |
| BMI (kg/m$^2$, mean ± SD) | 25.5 ± 2.8 | 28.6 ± 4.8 | 28.6 ± 4.6 | F = 2.737[a] | 0.077 |
| Hypertension, n (%) | 3 (21.4) | 2 (25.0) | 12 (57.1) | $\chi^2$ = 5.351[b] | 0.069 |
| DM, n (%) | 1 (7.1) | 1 (12.5) | 6 (28.6) | $\chi^2$ = 2.789[b] | 0.248 |
| Mean VAS pain score (mean ± SD) | 0.3 ± 0.6 | 1.5 ± 0.8** | 3.4 ± 0.7***### | F = 85.907[a] | < 0.0001 |
| Duration of pain (day, mean ± SD) | 75.1 ± 186.6 | 1665.0 ± 1200.0*** | 1060.0 ± 921.9** | F = 10.635[a] | < 0.0001 |
| Pain-relieving drugs used, n (%) | 5 (35.7) | 1 (12.5) | 7 (31.8) | $\chi^2$ = 1.488[b] | 0.475 |
| Opioid, n | 0 | 0 | 0 | | |
| NSAID, n | 2 | 1 | 6 | | |
| Acetaminophen, n | 3 | 0 | 1 | | |
| Operation (n) | Hernia (2) | OA hip replacement (2) | OA hip replacement (3) | | |
| | Hydrocele (1) | OA knee replacement (6) | OA knee replacement (17) | | |
| | Ureterscope (11) | | Knee arthroscopy (1) | | |

GU, genitourinary examination or operation. OA, osteoarthritis. VAS, Visual Analogue Scale. BMI, body mass index. DM, diabetes mellitus. NSAIDS, nonsteroidal anti-inflammatory drugs (etoricoxib, naproxen). Patients with GU diseases without obvious pain (VAS score 0–1 at rest) were included as a comparison (control) group. Patients with OA were divided into very mild pain (VAS score 0–2) and pain (VAS score $\geq$ 3) groups according to their level of pain. Data were presented as means ± standard deviation (SD), n, or n (%).

[a]One-way ANOVA with a Bonferroni post-hoc multiple comparison

**$p < 0.001$

***$p < 0.0001$ vs. GU control patients

###$p < 0.0001$ vs. OA patients with very mild pain.

[b]Analysis of $\chi^2$ test.

(mean duration of pain is 1,665 days), and OA patients with pain (mean duration of pain is 1,060 days) ($p < 0.0001$).

### 3.1 Plasma levels of TGF-β1 and BDNF were significantly down regulated in patients with chronic painful osteoarthritis

A multivariate analysis of covariance adjusted for the covariates (age, gender, BMI, hypertension, DM and pain-relieving drugs used) was used. And because of the non-normal distribution of these biological molecules, a non-parametric method (Kruskal- Wallis test) was used to analyze the data. OA patients with pain (VAS $\geq$ 3), but not those with VAS $\leq$ 2, had plasma TGF-β1 and BDNF levels that were significantly lower compared to the GU control patients (Table 2 and Fig 1A, $p < 0.05$).

### 3.2 The levels of TGF-β1 in CSF were significantly downregulated in the patients with chronic painful osteoarthritis

In the CSF, a multivariate analysis of covariance adjusted for the covariates (age, gender, BMI, hypertension, DM and pain-relieving drugs used) was used. And a non-parametric method (Kruskal- Wallis test) was also used to analyze the data. We demonstrated that the level of TGF-β1 (Table 2 and Fig 1B, $p < 0.05$) was significantly down regulated in the OA patients with pain compared to the GU control group. The levels of BDNF, TNF-α or IL-8 in CSF did not change in these patients with chronic OA compared to the GU control group (Table 2 and Fig 1B).

### 3.3 The levels of TGF-β1 in plasma and CSF were significantly correlated with the VAS scores

The Spearman correlation with control of age, gender, body mass index and pain drug showed that in patients with chronic OA, there were no significant correlations between age, gender,

**Table 2. Comparison of laboratory parameters in GU and chronic osteoarthritis patients.**

| | GU patients | Chronic OA patients | | MANCOVA | | Kruskal- Wallis Test |
|---|---|---|---|---|---|---|
| | VAS score 0–1 (control) | VAS score 0–2 (very mild pain) | VAS score $\geq$ 3 (pain) | F | p | p |
| N | 14 | 8 | 21 | | | |
| Plasma parameters (pg/mL) | | | | | | |
| TGF-β1 | 5108.3 ± 2094.6 | 1579.7 ± 661.1 | 739.3 ± 46.5*† | 4.790 | 0.015 | 0.035 |
| BDNF | 3952.0 ± 1106.5 | 2494.8 ± 1027.4 | 721.1 ± 211.6*†† | 4.896 | 0.014 | 0.001 |
| TNF-α | 1.87 ± 0.41 | 1.51 ± 0.58 | 1.21 ± 0.11 | 0.993 | 0.381 | 0.488 |
| IL-8 | 1.34 ± 0.32 | 1.39 ± 0.57 | 2.26 ± 0.45 | 0.286 | 0.753 | 0.251 |
| CSF parameters (pg/mL) | | | | | | |
| TGF-β1 | 38.41 ± 7.85 | 36.05 ± 10.63 | 12.16 ± 1.26*†# | 5.351 | 0.010 | 0.008 |
| BDNF | 12.11 ± 4.94 | 4.18 ± 2.97 | 1.74 ± 0.55 | 3.198 | 0.053 | 0.104 |
| TNF-α | 0.91 ± 0.34 | 0.27 ± 0.14 | 0.83 ± 0.30 | 0.175 | 0.840 | 0.395 |
| IL-8 | 74.28 ± 27.46 | 58.53 ± 28.39 | 68.05 ± 19.27 | 0.156 | 0.856 | 0.987 |

GU, genitourinary examination or operation. OA, osteoarthritis. VAS, Visual Analogue Scale. Patients with GU diseases without obvious pain (VAS score 0–1 at rest) were included as a comparison (control) group. Patients with OA were divided into very mild pain (VAS score 0–2) and pain (VAS score $\geq$ 3) groups according to their level of pain. Data were mean ± S.E. A multivariate analysis of covariance (MANCOVA) adjusted for covariates (age, gender, body mass index, hypertension, diabetes mellitus and pain-relieving drugs used) was used to analyze the data. Data are mean ± SE.

*$p < 0.05$ vs. GU control patients. And a non-parametric method (Kruskal- Wallis test) was used to analyze the data.

†$p < 0.05$

††$p < 0.01$ vs. GU control patients.

#$p < 0.05$ OA with very mild pain vs OA with pain.

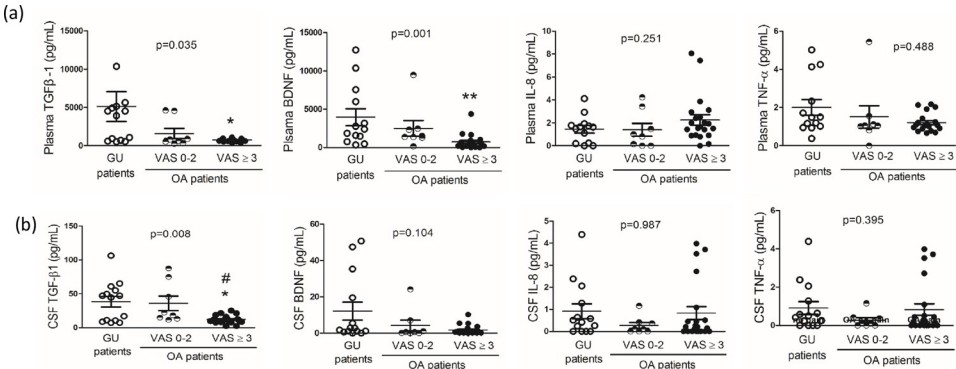

**Fig 1. The levels of cytokines and BDNF in plasma and cerebral fluid (CSF) in patients with genitourinary examination or operation (GU, control) and patients with osteoarthritis (OA).** OA patients were divided into very mild pain (Visual Analogue Scale, VAS score 0–2) and pain (VAS score ≥ 3) groups according to their level of pain. The non-parametric method (Kruskal- Wallis test) was used to analyze the data. Data are mean ± SE. *$p < 0.05$, ** $p < 0.01$ vs. GU control patients. #$p < 0.05$ OA with very mild pain vs OA with pain.

BMI, pain-relieving drugs used and the VAS scores. Their CSF TGF-β1 levels were significantly negatively correlated with their VAS pain scores (Table 3 and Fig 2, $p < 0.05$). However, there were no significant correlations between plasma and CSF BDNF, TNF-α and IL-8 levels and the VAS scores (Table 3).

## 4. Discussion

Up to the present time, treatment of pain has been a challenge because its pathogenesis and progression are very complex. Thus, an analysis of the underlying mechanism of pain in patients with osteoarthritis will help the treatment of chronic pain during the progression of osteoarthritis. This is the first study to simultaneously investigate the roles of cytokines and neurotrophic factors in human plasma and CSF for osteoarthritis patients with chronic pain. Our data demonstrated a significant decrease in the plasma and CSF levels of TGF-β1 in patients with osteoarthritis who were experiencing pain, but this was not found in those with very mild pain. In the osteoarthritis patients, the Spearman correlation analyses further revealed that the VAS pain scores were negatively correlated with the levels of TGF-β1 in CSF. However, there were no significant correlations between the pain scores and the levels of BDNF, TNF-α, and IL-8 in either the CSF or the plasma samples. These results indicate the important role of TGF-β1 in pain suffered by patients with chronic osteoarthritis.

**Table 3. Comparison of blood and CSF parameters with pain scores in patients with chronic osteoarthritis.**

| Plasma molecule | VAS score | | CSF molecule | VAS score | |
|---|---|---|---|---|---|
| | Spearman $r$ | $p$ | | Spearman $r$ | $p$ |
| Plasma TGF-β1 | 0.038 | 0.855 | CSFTGF-β1 | −0.498 | 0.011 |
| Plasma BDNF | −0.324 | 0.114 | CSFBDNF | 0.187 | 0.370 |
| Plasma TNF-α | −0.006 | 0.975 | CSFTNF-α | −0.016 | 0.938 |
| Plasma IL-8 | 0.151 | 0.471 | CSFIL-8 | −0.110 | 0.602 |

The correlation of plasma and cerebral spinal fluid (CSF) molecules with Visual Analogue Scale (VAS) scores in patients with chronic osteoarthritis (OA, n = 29). BDNF, brain derived neurotrophic factor. The Spearman correlation analysis with control of age, gender, body mass index and pain drug were used. $p < 0.05$ represented a significant correlation of variables with the VAS scores.

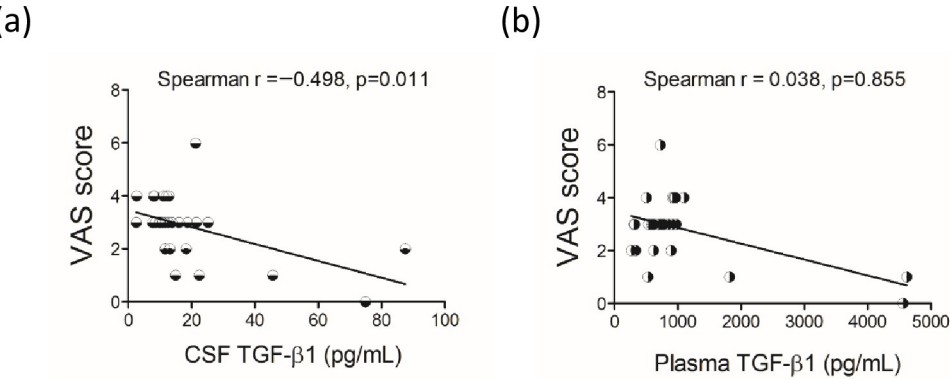

**Fig 2. Scatter plot of the Visual Analogue Scale (VAS) scores with the concentration of cerebral spinal fluid (CSF) and plasma TGF-β1 of patients with OA (n = 29).** Spearman correlation analyses with control of age, gender, body mass index and pain drug were used. $p < 0.05$ represented a significant correlation of variables with the VAS scores.

Several studies have reported that serum TNF-α is significantly higher in humans experiencing chronic pain, such as people with sciatica [10], fibromyalgia [20, 21], lumbar disc herniation [22], and low back pain [23]. Previous studies also have shown that the levels of inflammatory cytokines in the plasma of osteoarthritis patients [24] or synovial [25] samples are up-regulated. The downregulation of serum TNF-α has been correlated with improvements in the VAS scores of patients with osteoarthritis after treatment [5]. In this study, we did not find any significant differences in the levels of TNF-α and IL-8 in the plasma and CSF in osteoarthritis patients in the chronic pain group. However, our finding was partially supported by previous studies [21, 26–28] suggesting that the level of TNF-α in CSF do not increase in patients with complex regional pain syndrome [26] and post-traumatic osteoarthritis [28]. For patients with disc herniation and sciatica, the levels of TNF-α in their CSF and serum have also not been found to be significantly increased [27]. Our data suggest that the discrepancy in cytokines expression in different diseases may lead to variations in observations of pain. Here, it was assumed that inflammatory cytokines, which can be measured in the synovium, histological samples of osteoarthritis, blood and CSF, may differ depending on the stage of osteoarthritis. In this study, we collected patients with osteoarthritis and with chronic pain before they underwent joint replacement surgery. These patients may have suffered from pain for many years and are in the end stage of osteoarthritis. Our data represented that at the end stage of osteoarthritis, the levels of TNF-α and IL-8 in the plasma and CSF may not be the significant physiological indicators of pain.

Although the role of inflammatory cytokine expression in peripheral blood or the CNS remains controversial in terms of chronic pain, studies have suggested that intrathecal administration of anti-inflammatory cytokines can reverse chronic constriction injury-induced mechanical allodynia and thermal hyperalgesia in rats [29]. The significant decrease in the anti-inflammatory cytokine TGF-β1 in CSF and plasma found in the present study also hint at possible therapeutic targets for patients in pain. TGF-β1, a 25-kDa protein, expresses in different types of tissue and can be induced by activated glia cells [30]. It is classified as an anti-inflammatory cytokine [30]. The expression and production of TGF-β1 both in peripheral blood and brain have been found [30–33]. In blood, TGF-β1 has been found in high amounts within α granules of platelets and in platelet-rich plasma [31–33]. In peripheral tissue, platelets store large amounts of TGF-β1, which immediately after wounding facilitates the formation of a hemostatic plug [31] and accelerates wound healing *in vivo* [34]. In addition, platelet-rich plasma (PRP) has been intensively used for treatment of OA [35, 36], and patients with OA

treated with PRP showed improvement in terms of pain level and joint function compared to a control group [35]. Furthermore, numerous studies have shown that TGF-β1 is required for the formation of articular cartilage at the early stages of joint development. In mice, postnatal TGF-β1 signaling transduction molecule deletion in chondrocytes results in OA-like pathologies [37]. Activating the TGF-β1 signaling maintain the articular cartilage homeostasis and against the progression of OA [37]. Thus, the downregulation of plasma TGF-β1 in our patients with OA may decrease protection to the joint and accelerate disease progression. However, other investigations have indicated that TGF-β1 may, in fact, be a factor in joint destruction [38]. In an OA mouse model, high concentrations of active TGF–β1 in the subchondral bone initiated pathological changes in OA [38]. The conflicting roles of TGF-β1 signaling in joints in the development of OA may be due to the different stages of bone development and the concentration of TGF-β1 in local tissue.

In the central nervous system, astrocytes express and produce TGF-β1 [39]. TGF-β1 has dominant neuronal protective effects. It can promote the survival of dopaminergic neurons [40] and protect hippocampal neurons against transient global ischemia-induced damage [41]. In addition, TGF-β1 has been demonstrated to inhibit the intensity of pain in chronic constriction injury models [15, 42] and neuropathic pain [15] in rodents. Intrathecal injections of bone marrow stromal cells may increase TGF-β1 expression and inhibit neuropathic pain in mice [15]. The distinctive positive effects of TGF-β1 in terms of anti-inflammation, neuronal protection and the spinal sensitized pain clarifies its importance in chronic pain in central nervous system.

This is the first study to compare the levels of TGF-β1 in CSF and plasma in human subjects with chronic OA and with pain. Our data reveals that patients with end-stage OA experience long duration of pain (> 1000 days). In addition, patients with end-stage OA who experience very mild pain (VAS ≤ 2) at rest had a higher CSF TGF-β1 than those patients who experience pain at rest. Thus, we hypothesize that the levels of TGF-β1 in CSF in humans may modulate pain in OA subjects. Although the mechanism of TGF-β1 down-regulation in OA patients is still unclear, the findings of our study indicate that in patients with end stage of osteoarthritis chronic pain, the downregulation of TGF-β1 in CSF may be a significant indicator of the pain intensity associated with osteoarthritis.

With the exception of the downregulation of plasma and CSF TGF-β1, we also found a significant decrease in the level of plasma BDNF in patients with chronic pain. However, we did not find a significant downregulation of CSF BDNF in those experiencing chronic pain. In the CNS, BDNF is produced from neurons and the glia cells [43, 44] and also serves as a protective neuronal factor. BDNF has quite different effects from those of TGF-β1 in pain studies [17, 18, 45, 46]. In rodent inflammatory or neuropathic pain models, BDNF was found to be potentiated and involved in the generation of pain [17, 18]. Serum levels of BDNF were found to be increased in older women with low back pain [45], and plasma levels of BDNF were positively correlated with the severity of pain in women with pelvic pain [46]. However, for patients with post-herpetic neuralgia, lower levels of BDNF in CSF have been reported [47]. Thus, the roles of BDNF in pain still require further study.

There are some limitations in this study. First of all, there was no CSF from the healthy subjects. We collected it from a disease reference group without obvious pain (VAS score 0–1). We did not exclude the molecular differences in circulation or CSF between unhealthy and healthy control subjects. Second, the use of analgesics (acetaminophen and NSAIDs) may impact the expression of cytokines. Although we attempted to exclude their impact through the use of a partial correlation with adjusted the used of pain-relieving drugs, further studies still need to be conducted to confirm the effects of analgesics on cytokines in CSF and blood in patients with pain. Third, although the ELISA kits were validated to measure these molecules

in human plasma and supernatants of cell culture. And the recombined human IL-8, TNF-α, BDNF and TGF-β1 diluted in buffer were used as a calibration. However, this is the first time ELISA kits is used to detect these molecules in CSF, and the interpretation of CSF data should be more cautious.

## 5. Conclusions

Our findings revealed that the levels of TGF-β1 in plasma and CSF may be biological indicators in patients with end stage osteoarthritis experiencing chronic pain. Therapeutics that promote TGF-β1 expression may serve as a therapy for pain control in patients with osteoarthritis. The levels of BDNF, TNF-α, or IL8, were not shown to be suitable biological indicators of chronic pain.

## Acknowledgments

Dr. Ru-Band Lu advised on data analysis. We sincerely thank our colleagues at NCKU Hospital for their assistance in sample collection.

## Author Contributions

**Formal analysis:** Yen-Chin Liu.

**Funding acquisition:** Yen-Chin Liu, Shiou-Lan Chen.

**Investigation:** Hung-Tsung Hsiao, Jeffrey Chi-Fei Wang, Tzu-Cheng Wen, Shiou-Lan Chen.

**Methodology:** Yen-Chin Liu.

**Project administration:** Yen-Chin Liu, Shiou-Lan Chen.

**Writing – original draft:** Yen-Chin Liu, Shiou-Lan Chen.

**Writing – review & editing:** Yen-Chin Liu, Hung-Tsung Hsiao, Jeffrey Chi-Fei Wang, Tzu-Cheng Wen, Shiou-Lan Chen.

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
