## [Decision Letter · Decision Letter 0]

22 Jun 2021

PONE-D-20-38558

TGF-β1 in plasma and cerebrospinal fluid can be used as a biological indicator of chronic pain sensation in patients with osteoarthritis

PLOS ONE

Dear Dr. Chen,

Thank you for submitting your manuscript to PLOS ONE. After careful consideration, we feel that it has merit but does not fully meet PLOS ONE’s publication criteria as it currently stands. Therefore, we invite you to submit a revised version of the manuscript that addresses the points raised during the review process.

We look forward to receiving your revised manuscript.

Kind regards,

Firas H Kobeissy, PhD

Academic Editor

PLOS ONE

Journal Requirements:

 No.

This study was supported in part by grant NCKUH-10509003 from National Cheng Kung University Hospital, Taiwan and 105-2314-B-006-007-, 106-2314-B-006-026-MY2 (to YCL), 105-2628-B-037 -003 -MY3 (to SLC), from the Ministry of Science and Technology, Taiwan.

This study was supported in part by grant NCKUH-10509003 from National Cheng Kung University Hospital, Taiwan and 105-2314-B-006-007-, 106-2314-B-006-026-MY2 (to YCL), 105-2628-B-037 -003 -MY3 (to SLC), from the Ministry of Science and Technology, Taiwan.

Reviewers' comments:

Reviewer's Responses to Questions

**Comments to the Author**

1. Is the manuscript technically sound, and do the data support the conclusions?

Reviewer #1: Partly

Reviewer #2: Partly

2. Has the statistical analysis been performed appropriately and rigorously? 

Reviewer #1: No

Reviewer #2: Yes

3. Have the authors made all data underlying the findings in their manuscript fully available?

Reviewer #1: No

Reviewer #2: No

4. Is the manuscript presented in an intelligible fashion and written in standard English?

Reviewer #1: No

Reviewer #2: No

5. Review Comments to the Author

Reviewer #1: This interesting manuscript presents data on cytokine-immunoreactivity in paired plasma and CSF from people with OA and urological controls. The findings are of interest, not least because they are not what one might expect. As such, the manuscript raises more questions than it answers, although much of this might be addressed by more detailed presentation of methods and data.

Specific points:

What was the lead question and recall timeframe for the pain VAS?

Please refer to molecule detection by ELISA to indicate that what was measured was immunoreactivity (e.g. BDNF-LI), or otherwise provide evidence of assay validation with this sample type that confirms molecular identity with e.g. BDNF.

Please specify diagnoses in control group in more detail, as, in the absence of a healthy matched control group, it might be that controls are abnormal, rather than the patients with OA. How do the authors explain that 5 in the control group were using pain-relieving medications if they did not have pain?

The mean VAS for those with OA pain (3.4) is very low (Table 1) considering that all these people had at least VAS=3. What was the range/distribution of VAS scores in the groups?

Were cytokine levels and VAS scores normally distributed? Cytokine levels will usually be positively skewed, requiring logarithmic transformation prior to parametric analysis. It seems that the range of VAS scores is very low (mild pain). Descriptives (table 1) should normally be presented with SD or interquartile range rather than SEM, to enable the reader to gain an impression of the distribution of data, rather than just the precision of the mean estimate. Additionally, the regression analyses might be clearer if presented through scatterplots, to demonstrate that the significant Pearson correlation coefficients are not driven by a small number of extreme values.

Please include diagnostic group as a covariate (ie control or OA) in regression analyses in case observed differences are attributable to other aspects of diagnosis rather than pain.

The origin of TGF beta in the current population is unclear, however, differences in plasma levels as well as CSF levels might well point to a peripheral source. TGF beta has been associated with OA pathology and it would be important to relate the current findings to previous data on circulating TGF beta in OA. Furthermore, please describe the `stage’ of OA in more detail (mentioned in discussion as end stage). Given that knee structural changes in OA are also associated with pain, might TGF beta be a marker for structural severity rather than pain severity?

The authors should be commended for not emphasising conclusions from their negative findings, which might have resulted from power limitations.

Pain is, by definition, a sensation. Please avoid the term `pain sensation’ which is easily misread for `pain sensitisation’ which is not studied here.

Reviewer #2: PLos ONE

TGF-β1 in plasma and cerebrospinal fluid can be used as a biological indicator of chronic pain sensation in patients with osteoarthritis.

Yen-Chin Liu, Hung-Tsung Hsiao, Jeffrey Chi-Fei Wang, Tzu-Cheng Wen, Shiou-Lan Chen.

Cytokines can affect pain in osteoarthritis (OA), as well as in many other conditions, in various ways e.g. by enhancing or depressing inflammatory, and various other tissue changes, but also by direct effects on the nervous system. The authors present their data on the plasma and CSF levels of 4 cytokines in patients with OA of the hip, and/or knee. The cytokines they studied were TGF-β1, BDNF, TNF-α, and IL-8. Plasma and CSF samples were obtained from patients scheduled to have spinal anesthesia for surgery. Plasma and CSF samples were also obtained from patients with genitourinary system disease (GU), scheduled for surgery under spinal anesthesia. The samples were assessed for levels of the above 4 cytokines. The authors compared the levels of each of the 4 cytokines between the group of OA patients with little or no pain, and OA patients with more severe pain. The authors also compared cytokine levels between OA patients and GU patients. Their findings showed a significant difference in TGF-β1 levels in OA patients with severe pain and OA patients with mild or no pain; CSF and plasma TGF-β1 levels were significantly lower in the OA patients with severe pain compared to those levels in OA patients with no or mild pain. Similarly, the OA patients with severe pain also had significantly lower TGF-β1 levels (CSF and plasma) than the GU patients. BDNF plasma levels were lower in patients with OA associated with severe pain, but there was no significant lowering of BDNF levels in CSF. The lower plasma BDNF levels represented an unexpected finding for which there does not appear to be a good explanation. There were no significant differences between the patients with severe OA, mild OA, and GU with respect to IL-8 and TNF-α levels in either plasma or CSF. In OA patients with severe pain there was a significant negative correlation between pain severity (assessed by VAS) and TGF-β1 levels (both plasma and CSF).

Assessment.

Major Points

1.These results are interesting, but it is uncertain whether they are generalizable because of some questions with respect to choices in the patient populations and their clinical assessments.

*The authors included 36 patients with OA; they excluded 5 for “a chronic pain history with acute pain surgery”. What does this mean?

*Two more were excluded because they had neural disease “when they received spinal anesthesia for the osteoarthritis of knee or hip surgery” What does this mean?

*It is not clear what surgery the other 29 OA patients had. I assume that it would have been either knee or hip arthroplasties but this has to be specified.

*If the above is correct, it appears that such OA patients differ from the usual population selected for arthroplasties in Europe or North America where such operations are performed in a somewhat older population with a male predominance.

*Eight out of the 26 patients with OA had a VAS of 0-2. Patients with OA of hips or knees proceeding

to an arthroplasty have severe pain. A VAS of 0-2 would be unusual in such patients.

2. The VAS was assessed at the time the patients entered the hospital. I am concerned about the validity of such an assessment. An average VAS for the preceding week would be more informative.

3. The GU comparison group was selected as “having no pain”. Yet in Table 1 it appears that 5/14 of these patients were taking “pain relief drugs”. Somewhat surprisingly, only 7/21 OA patients with VAS scores of > were taking such medications.

4. The low levels of TGF-β1 in patients with more severe pain are an interesting finding. The authors should discuss the possible reasons for this. Could this be due to inadequate production or excessive consumption? What do animal experiments suggest?

Minor Points

1. The paper requires extensive corrections with respect to the English language. Examples : Page 2 (Introduction) line 2 : “The persisted chronic pain has been considered”; Page 11, lines 240-241 ( first and second from the top) in the “Discussion” part : “Our data suggest that then discrepancy of cytokines expression in different disease may provide the multiple aspect of observation in pain sensation”.

2. Each table should be presented on a separate page.

6. PLOS authors have the option to publish the peer review history of their article (what does this mean?). If published, this will include your full peer review and any attached files.

Reviewer #1: No

Reviewer #2: No

---

## [Author Response · Author response to Decision Letter 0]

5 Aug 2021

Reviewers' comments:

Reviewer #1: This interesting manuscript presents data on cytokine-immunoreactivity in paired plasma and CSF from people with OA and urological controls. The findings are of interest, not least because they are not what one might expect. As such, the manuscript raises more questions than it answers, although much of this might be addressed by more detailed presentation of methods and data.

Response: We thank the reviewer for the positive words. As recommended by the reviewer, we have added the revision in the manuscript.

Specific points:

What was the lead question and recall timeframe for the pain VAS? 

Response: This is a cross-sectional study of CSF and plasma molecules in human subjects with chronic pain. We evaluated the intensity of pain before surgery. So even the GU patients were evaluated. About the OA patients, all patients fit the OA X ray criteria (including joint space loss and osteophyte formation) and passed another objective orthpedicst's check, their chronic OA diagnosis was confirmed. About patients' VAS recall timeframe, we asked patient recall their VAS when admission. We also added the duration of pain to confirm their chronic pain situation. The related descriptions were revised in method (Page 6, last paragraph), Table 1 and results (Page 8, last paragraph).

Please refer to molecule detection by ELISA to indicate that what was measured was immunoreactivity (e.g. BDNF-LI), or otherwise provide evidence of assay validation with this sample type that confirms molecular identity with e.g. BDNF. 

Response: As we mentioned in method, the commercial ELISA kit (Quantikine Human Cytokine Kit; R&D Systems, Minneapolis, MN, USA) were used for the measurement of protein levels of TNF-α, IL-8, TGF-�1 and mature form of BDNF (Page 7, last paragraph).

Please specify diagnoses in control group in more detail, as, in the absence of a healthy matched control group, it might be that controls are abnormal, rather than the patients with OA. How do the authors explain that 5 in the control group were using pain-relieving medications if they did not have pain? 

Response: We agree with the comment, thus this point was described in the limitation of this study (Page 21, lines 326-328). We also revised our Table 1 to present more details of each groups. We used the GU patients with no pain as a control group and presented no significant differences of age, gender, body weight, or body mass index (BMI) between-groups. However, the significant difference of VAS scores and pain duration were found between groups (Table 1).

 About the 5 GU patients that using pain-relieving medications that we founded from their outpatient clinic record, we think it is the ethic issue that doctors prescribed the pain -relieving medication because of the complain of pain of patients in the outpatient clinic. Thus, only weak pain killers were used in this group. And as we revised in Table 1 that 3 of those GU patients used acetaminophen and 2 of those GU patients used NSAIDs (etoricoxib, naproxen) which were adjusted as covariates in Table 2 and figure 1. We also described this point in the limitation of this study (Page 21, lines 328-332).

The mean VAS for those with OA pain (3.4) is very low (Table 1) considering that all these people had at least VAS=3. What was the range/distribution of VAS scores in the groups? 

Response: In this study, we found that OA patients with very mild pain (n=8), their VAS score are 0 to 2. And in OA subjects with chronic pain (n=21), the range of VAS scores are 3 to 6. 

Were cytokine levels and VAS scores normally distributed? Cytokine levels will usually be positively skewed, requiring logarithmic transformation prior to parametric analysis. It seems that the range of VAS scores is very low (mild pain). Descriptives (table 1) should normally be presented with SD or interquartile range rather than SEM, to enable the reader to gain an impression of the distribution of data, rather than just the precision of the mean estimate. Additionally, the regression analyses might be clearer if presented through scatterplots, to demonstrate that the significant Pearson correlation coefficients are not driven by a small number of extreme values.

Response: We had revised the Table 1 as the suggestion. And we thanks the reviewer's comments that the distribution of cytokine levels are indeed positively skewed. Thus, we also transformed the levels of cytokines logarithmically prior to the analysis of Pearson correlation in Table 3 (Page 16). The scatterplots of VAS with log CSF TGF-β1 and log plasma TGF-β1 are also shown in Fig 2 (Page 17).

Please include diagnostic group as a covariate (ie control or OA) in regression analyses in case observed differences are attributable to other aspects of diagnosis rather than pain.

Response: In Table 3 (Page 16), we only included the patients with same diagnosis (OA patients, n=29). And we transformed the levels of cytokines logarithmically prior to the analysis of Pearson correlation. We also added the scatterplots of VAS with log CSF TGF-β1 and log plasma TGF-β1 are also shown in Fig 2 (Page 17).

The origin of TGF beta in the current population is unclear, however, differences in plasma levels as well as CSF levels might well point to a peripheral source. TGF beta has been associated with OA pathology and it would be important to relate the current findings to previous data on circulating TGF beta in OA. 

Response: We thank the reviewer for the excellent comments. We had added the description of origin of TGF beta in blood and brain. And the possible relationship of TGF-β1 in OA (Page 19, last paragraph and Page 20, lines 299-315)

Furthermore, please describe the `stage’ of OA in more detail (mentioned in discussion as end stage). Given that knee structural changes in OA are also associated with pain, might TGF beta be a marker for structural severity rather than pain severity? 

Response: Thanks the comments. We added the describe in result (Page 8, lines 172-174) that the definition of end-stage OA was that X-ray presented articular cartilage wears away almost completely and bone on bone contact occurs and passed another objective orthpedicst’s check. We also added the duration of pain of each groups in Table 1.

In addition, we agree with the comments of reviewer, that the levels of plasma TGF-β1 may as a marker for structural severity of OA. However, in this study we collect the CSF TGF-β1 that may explain the intensity of central pain in OA subjects. However, the expression of TGF-β1 in different stage of OA need more study in the future. 

The authors should be commended for not emphasising conclusions from their negative findings, which might have resulted from power limitations.

Response: Thank you for this comment.

Pain is, by definition, a sensation. Please avoid the term `pain sensation’ which is easily misread for `pain sensitisation’ which is not studied here. 

Response: Thank you for this comment. In this revision, we revised VAS as the measurement of pain intensity. We revised most of the “pain sensation” description as “pain intensity” or “the intensity of pain”. 

Reviewer #2: PLos ONE

Cytokines can affect pain in osteoarthritis (OA), as well as in many other conditions, in various ways e.g. by enhancing or depressing inflammatory, and various other tissue changes, but also by direct effects on the nervous system. The authors present their data on the plasma and CSF levels of 4 cytokines in patients with OA of the hip, and/or knee. The cytokines they studied were TGF-β1, BDNF, TNF-α, and IL-8. Plasma and CSF samples were obtained from patients scheduled to have spinal anesthesia for surgery. Plasma and CSF samples were also obtained from patients with genitourinary system disease (GU), scheduled for surgery under spinal anesthesia. The samples were assessed for levels of the above 4 cytokines. The authors compared the levels of each of the 4 cytokines between the group of OA patients with little or no pain, and OA patients with more severe pain. The authors also compared cytokine levels between OA patients and GU patients. Their findings showed a significant difference in TGF-β1 levels in OA patients with severe pain and OA patients with mild or no pain; CSF and plasma TGF-β1 levels were significantly lower in the OA patients with severe pain compared to those levels in OA patients with no or mild pain. Similarly, the OA patients with severe pain also had significantly lower TGF-β1 levels (CSF and plasma) than the GU patients. BDNF plasma levels were lower in patients with OA associated with severe pain, but there was no significant lowering of BDNF levels in CSF. The lower plasma BDNF levels represented an unexpected finding for which there does not appear to be a good explanation. There were no significant differences between the patients with severe OA, mild OA, and GU with respect to IL-8 and TNF-α levels in either plasma or CSF. In OA patients with severe pain there was a significant negative correlation between pain severity (assessed by VAS) and TGF-β1 levels (both plasma and CSF).

Assessment.

Major Points

1.These results are interesting, but it is uncertain whether they are generalizable because of some questions with respect to choices in the patient populations and their clinical assessments.

*The authors included 36 patients with OA; they excluded 5 for “a chronic pain history with acute pain surgery”. What does this mean? 

Response: Sorry for the misunderstanding. Actually, we collected 5 OA patients with low-leg trauma and with acute pain. Thus, these patients with chronic and acute pain were excluded. We deleted this sentence (Page 8) for possible misunderstanding.

*Two more were excluded because they had neural disease “when they received spinal anesthesia for the osteoarthritis of knee or hip surgery” What does this mean? 

Response: We deleted this sentence (Page 8). Actually, we had mentioned in the exclusion criteria 2) we excluded those patients having pre-existing neurological diseases. The pre-existing neurological disease may have influence on CSF data. 

*It is not clear what surgery the other 29 OA patients had. I assume that it would have been either knee or hip arthroplasties but this has to be specified. 

Response: Thank you for your comment. We revised the Table 1 and presented the surgery type which was OA hip (total hip replacement), OA knee (total knee replacement) and knee arthroscopy. 

*If the above is correct, it appears that such OA patients differ from the usual population selected for arthroplasties in Europe or North America where such operations are performed in a somewhat older population with a male predominance. 

Response: Thank you for your comment. We agree that generally, male may receive more OA surgery than female. However, some data still claim that OA is more common in women than in men. Although we have the limitation in the selection of case, however, the comparison of VAS scores with CSF and plasma molecules may provide a reference in the observation of OA with chronic pain.

*Eight out of the 26 patients with OA had a VAS of 0-2. Patients with OA of hips or knees proceeding to an arthroplasty have severe pain. A VAS of 0-2 would be unusual in such patients. 

Response: Thank you for your comment. We agree that OA surgery caused a lot of pain. However, we had described clearly that patients’ VAS was recorded when they administered to the hospital before surgery, not after surgery. And patients were asked to present their VAS before operation. It is also reasonable that OA patients have low VAS once they reduced activity or taking drugs. We thought their low VAS may present some clinical meanings and separated as another groups 

2. The VAS was assessed at the time the patients entered the hospital. I am concerned about the validity of such an assessment. An average VAS for the preceding week would be more informative. 

Response: Thank you for your comment. We agreed that the VAS validity is an important issue. However, we also concerned the recall validity if we asked patient recall the preceding week. Furthermore, there is no data showed that recalling preceding week VAS is a better presentation than VAS at administration. So we used the VAS at administration as grouping category and we also noticed that those with low VAS at administration also used less pain drug than those with high VAS at administration which also proved that our grouping using VAS at administration is reasonable.

3. The GU comparison group was selected as “having no pain”. Yet in Table 1 it appears that 5/14 of these patients were taking “pain relief drugs”. Somewhat surprisingly, only 7/21 OA patients with VAS scores of > were taking such medications. 

Response: Thank you for your comment. About the 5 GU patients that using pain killers, that we founded from their outpatient clinic record, we think it is the ethic issue that doctors prescribed the pain killer because of the complain of pain of patients in the outpatient clinic. Thus, only weak pain killers were used in this group. And as we presented in the revised Table 1 that 3 of those GU patients used acetaminophen and 2 of those GU patients used NSAIDs (etoricoxib, naproxen) which were adjusted as covariates in Table 2 and figure 1. We also described this point in the limitation of this study (Page 21, lines 328-332)

For OA patients, we still find that less pain group patient used less pain drugs (1/8 vs.7/21) We also think that, in our hospital, many patients receive surgery because they don’t want taking pain drugs because of many reasons (afraid GI/renal side effect, already taken many drug, still pain even taking drug….). So even the surgeon prescribed less drugs if patients claimed they don’t want take medicine. The surgeon just arrange operation as soon as possible. So there are less ratio for patients receiving drugs.

4. The low levels of TGF-β1 in patients with more severe pain are an interesting finding. The authors should discuss the possible reasons for this. Could this be due to inadequate production or excessive consumption? What do animal experiments suggest?

Response: This is a first study that compared the levels of TGF-β1, BDNF and cytokines in CSF and plasma of human subjects with no pain, very mild pain and chronic pain condition. As we discussed in page 19-20 that TGF-β1 may involve in the articular cartilage homeostasis and inhibit pain sensation in rodents. Thus we hypothesize that the levels of TGF-β1 in CSF/plasma in humans may associated with the progression of disease and modulate pain. Although the mechanism of TGF-β1 down-regulation in OA patients is still unclear. However, our study represented that patients with end stage of osteoarthritis and with chronic pain, the downregulation of TGF-β1 in CSF may be a significant indicator of the pain intensity associated with osteoarthritis. The related description of TGF-β1 with OA progression and chronic pain were added in Page 19 last paragraph and Page 20 lines 299-315.

Minor Points

1. The paper requires extensive corrections with respect to the English language. Examples : Page 2 (Introduction) line 2 : “The persisted chronic pain has been considered”; Page 11, lines 240-241 ( first and second from the top) in the “Discussion” part : “Our data suggest that then discrepancy of cytokines expression in different disease may provide the multiple aspect of observation in pain sensation”.

Response: Thank you for your comment. We revised this newest version again and make it easy to read. We also corrected the grammar errors with the native speaker's help.

2. Each table should be presented on a separate page.

Response: We had presented each table on a separate page.

---

## [Decision Letter · Decision Letter 1]

13 Sep 2021

PONE-D-20-38558R1TGF-β1 in plasma and cerebrospinal fluid can be used as a biological indicator of chronic pain in patients with osteoarthritisPLOS ONE

Dear Dr. Chen,

Thank you for submitting your manuscript to PLOS ONE. After careful consideration, we feel that it has merit but does not fully meet PLOS ONE’s publication criteria as it currently stands. Therefore, we invite you to submit a revised version of the manuscript that addresses the points raised during the review process.

We look forward to receiving your revised manuscript.

Kind regards,

Firas H Kobeissy, PhD

Academic Editor

PLOS ONE

Journal Requirements:

Reviewers' comments:

Reviewer's Responses to Questions

**Comments to the Author**

1. If the authors have adequately addressed your comments raised in a previous round of review and you feel that this manuscript is now acceptable for publication, you may indicate that here to bypass the “Comments to the Author” section, enter your conflict of interest statement in the “Confidential to Editor” section, and submit your "Accept" recommendation.

Reviewer #1: (No Response)

Reviewer #2: All comments have been addressed

2. Is the manuscript technically sound, and do the data support the conclusions?

Reviewer #1: Partly

Reviewer #2: Yes

3. Has the statistical analysis been performed appropriately and rigorously? 

Reviewer #1: No

Reviewer #2: Yes

4. Have the authors made all data underlying the findings in their manuscript fully available?

Reviewer #1: No

Reviewer #2: Yes

5. Is the manuscript presented in an intelligible fashion and written in standard English?

Reviewer #1: No

Reviewer #2: Yes

6. Review Comments to the Author

Reviewer #1: I appreciate the authors’ efforts in revising their manuscript, which goes some way to address my original concerns. I suspect that my original comments (and those of the second reviewer where similar) might have been incompletely understood by the authors and I attempt to explain my residual concerns more clearly below.

What was the lead question and recall timeframe for the pain VAS? (?p6) I note also that this point was raised by the second reviewer, who made additional pertinent points. Having read the authors’ response, I think that I might understand the source of apparent confusion and discrepancies, but this is not yet clear to the general reader of the manuscript. Am I correct to think that participants were recruited on the basis of clinical evidence of chronic pain (undergoing arthroplasty), but that the pain VAS used in analyses asked about `recent pain’. As in my original comments, please give the exact text and anchors for the VAS used. I expect this was not in English language, and so the exact text in the original language, plus a verbatim translation might suffice. It makes a very big difference what time-frame defined `recent’ if the original text used the word `recent’. Does `recent’ mean since admission, over the past few minutes while you have not been standing, or over the past few months? I suspect from the results that participants rated their `current’ pain whilst not mobilising or standing. Our experience also is that patients listed for arthroplasty might not have severe pain at rest, but as soon as they try to do something, then their pain becomes severe. Chronic pain (e.g. as measured by WOMAC or clinically used VAS scores) typically refers to pain over the past week; either average pain, or worst pain, thereby reflecting the intermittent nature often of OA pain. On reflection, I can understand that `current pain’ might be the most appropriate index here, because molecules in CSF might change over short time periods, and the authors might wish to measure the pain experienced at a time point closest to biosample collection. However, if this is the case, they might wish to speculate as to how relevant this might be to clinical pain. At least, they should explain the apparent inconsistency commented on by both reviewers that people with very low pain scores were undergoing joint surgery. Also I note that reviewer 2, like myself, commented on the GU comparison group was selected as “having no pain”; so am I correct to understand that they had no `current pain on admission to hospital’ but that they might have had chronic pain prior to admission? This again should be clarified, and might be part of the same explanation as the clarification of what the VAS score represents.

The authors have chosen to not `indicate that what was measured was immunoreactivity (e.g. BDNF-LI)’. I do not feels strongly about this, but the fact that ELISA measurements, especially competitive ELISAs, might measure factors other than the specific molecular species targeted, particularly when they are applied to biofluids in which the manufacturer and investigator have not validated them. I would still suggest that the authors provide data on validation in plasma and CSF, rathe than assume that these kits do what the manufacturers say they do when they sell them for profit.

Please provide statistical evidence that log transformed data are normally distributed. If, as stated in the authors’ response, untransformed data were significantly positively skewed, then please use log transformed (or otherwise normalised) data for all parametric analyses, including between group comparisons. If log transformation converts non-normal data to normal data, then please only present analyses on log transformed data (e.g. in Table 2). In other words, only present the statistically appropriate analyses.

I’m afraid that I don’t understand the data in the statement; ` The mean duration of pain was 75.1, 1,665, and 1,060 days with no pain, OA patients with very mild pain, and OA patients experiencing pain according to the recall of the patients, respectively (p < 0.0001).’ Are the authors really stating that participants were asked how long they had had pain to the nearest 0.1 days??? Given my above comments about variability of pain in OA, this seems meaningless if I am interpreting correctly. What is the method for this?

Reviewer #2: My concerns have been answered. The authors' findings are interesting although their significance is uncertain in view of conflicting evidence in thre literature on the roles of TGFbeta1 and BDNF.

7. PLOS authors have the option to publish the peer review history of their article (what does this mean?). If published, this will include your full peer review and any attached files.

Reviewer #1: **Yes: **David Andrew Walsh

Reviewer #2: No

---

## [Author Response · Author response to Decision Letter 1]

26 Oct 2021

Reviewers' comments:

Response: We thank the reviewers for the positive words. As recommended by the reviewers, we have added the revision in the manuscript.

Reviewer #1: I appreciate the authors’ efforts in revising their manuscript, which goes some way to address my original concerns. I suspect that my original comments (and those of the second reviewer where similar) might have been incompletely understood by the authors and I attempt to explain my residual concerns more clearly below.

What was the lead question and recall timeframe for the pain VAS? (?p6) I note also that this point was raised by the second reviewer, who made additional pertinent points. Having read the authors’ response, I think that I might understand the source of apparent confusion and discrepancies, but this is not yet clear to the general reader of the manuscript. Am I correct to think that participants were recruited on the basis of clinical evidence of chronic pain (undergoing arthroplasty), but that the pain VAS used in analyses asked about `recent pain’. As in my original comments, please give the exact text and anchors for the VAS used. I expect this was not in English language, and so the exact text in the original language, plus a verbatim translation might suffice. It makes a very big difference what time-frame defined `recent’ if the original text used the word `recent’. Does `recent’ mean since admission, over the past few minutes while you have not been standing, or over the past few months? I suspect from the results that participants rated their `current’ pain whilst not mobilising or standing. Our experience also is that patients listed for arthroplasty might not have severe pain at rest, but as soon as they try to do something, then their pain becomes severe. Chronic pain (e.g. as measured by WOMAC or clinically used VAS scores) typically refers to pain over the past week; either average pain, or worst pain, thereby reflecting the intermittent nature often of OA pain. On reflection, I can understand that `current pain’ might be the most appropriate index here, because molecules in CSF might change over short time periods, and the authors might wish to measure the pain experienced at a time point closest to biosample collection. 

Response: Thank you for your comments. Correctly, all patients were asked to evaluate their current pain intensity using the Visual Analogue Scale (VAS, ranged from 0-10, no moving or standing), based on their native language Mandarin or Taiwanese a day before the operation and biosample collection. All of these were revised in method (Page 6, last paragraph) and results (Page 8, last paragraph).

However, if this is the case, they might wish to speculate as to how relevant this might be to clinical pain. At least, they should explain the apparent inconsistency commented on by both reviewers that people with very low pain scores were undergoing joint surgery. 

Response: Thank you for your comments. And, as you recognized, our data support that patients with end-stage OA experienced with chronic pain (> 1000 days, Table 1). In addition, our data also revealed that some patients with chronic OA do not feel obvious pain when resting (VAS ≤ 2, in OA with very mild pain group). And this patients (OA with very mild pain) at rest had higher CSF TGF-β1 than those OA patients with pain. Thus, we hypothesize that the levels of TGF-β1 in CSF in humans may modulate pain in OA subjects. All of these were revised in result (Page 8, last paragraph; Page 9, last paragraph) and discussion (Page 20, line 328-330).

 

Also I note that reviewer 2, like myself, commented on the GU comparison group was selected as “having no pain”; so am I correct to understand that they had no `current pain on admission to hospital’ but that they might have had chronic pain prior to admission? This again should be clarified, and might be part of the same explanation as the clarification of what the VAS score represents. 

Response: Thank you for your comments. We apology that we did not described the definition and condition of the GU-control group clearly. In this study, patients with GU diseases without obvious current pain at rest (VAS score 0-1, mean VAS score is 0.3 ± 0.6 in GU control patients, Table 1) before surgery were included as a comparison (control) group. Furthermore, to access that they might have had chronic pain, we also checked the experience of pain of GU patients according to their recollection (GU control, mean duration of pain, 75.1 days in Table 1). All of these were revised in method (Page 6, lines 116-117) and result (Page 9).

The authors have chosen to not `indicate that what was measured was immunoreactivity (e.g. BDNF-LI)’. I do not feels strongly about this, but the fact that ELISA measurements, especially competitive ELISAs, might measure factors other than the specific molecular species targeted, particularly when they are applied to biofluids in which the manufacturer and investigator have not validated them. I would still suggest that the authors provide data on validation in plasma and CSF, rathe than assume that these kits do what the manufacturers say they do when they sell them for profit. 

Response: We agree that although the ELISA kit were validated to measure these molecules in human plasma and supernatants of cell culture. And the recombined human IL-8, TNF-α, BDNF and TGF-β1 diluted in buffer were used as a calibration. However, this is the first time that using of ELISA kits to detect these molecules in CSF, and the interpretation of CSF data should be more cautious. We have added and revised such description in method (Page 7, last paragraph; Page 8, 1st paragraph) and limitations (Page 21).

Please provide statistical evidence that log transformed data are normally distributed. If, as stated in the authors’ response, untransformed data were significantly positively skewed, then please use log transformed (or otherwise normalised) data for all parametric analyses, including between group comparisons. If log transformation converts non-normal data to normal data, then please only present analyses on log transformed data (e.g. in Table 2). In other words, only present the statistically appropriate analyses. 

Response: Thanks for the suggestion of statistics. Because of the log transformed data are still not normally distributed. For the biological molecules are often non-normality, thus, a non-parametric statistical test (Kruskal-Wallis test) was added to analysis the original data (Table 2). To determine the correlation between the VAS pain scores and the plasma and CSF molecule, the Spearman correlations with control of age, gender, body mass index and pain drug were performed (Table 3). All of these were added in method (Page 8, 2nd paragraph), Table 2, Table 3 and results (Page 12, Page 15).

I’m afraid that I don’t understand the data in the statement; ` The mean duration of pain was 75.1, 1,665, and 1,060 days with no pain, OA patients with very mild pain, and OA patients experiencing pain according to the recall of the patients, respectively (p < 0.0001).’ Are the authors really stating that participants were asked how long they had had pain to the nearest 0.1 days??? Given my above comments about variability of pain in OA, this seems meaningless if I am interpreting correctly. What is the method for this? 

Response: We apology that we did not clearly described the definition of this data. In this study, to confirm the situation of chronic pain, the duration of pain (days) was collected. Patients were asked to recall how long the pain or discomfort caused by the disease has persisted. All of these were added and revised in Table 1, method (Page 7, 1st paragraph) and in result (Page 9, last paragraph).

Reviewer #2: My concerns have been answered. The authors' findings are interesting although their significance is uncertain in view of conflicting evidence in thre literature on the roles of TGFbeta1 and BDNF.

Response: Thank you for you insightful feedback and comments.

---

## [Decision Letter · Decision Letter 2]

19 Dec 2021

TGF-β1 in plasma and cerebrospinal fluid can be used as a biological indicator of chronic pain in patients with osteoarthritis

PONE-D-20-38558R2

Dear Dr. Chen,

We’re pleased to inform you that your manuscript has been judged scientifically suitable for publication and will be formally accepted for publication once it meets all outstanding technical requirements.

Kind regards,

Firas H Kobeissy, PhD

Academic Editor

PLOS ONE

Additional Editor Comments (optional):

Reviewers' comments:

Reviewer's Responses to Questions

**Comments to the Author**

1. If the authors have adequately addressed your comments raised in a previous round of review and you feel that this manuscript is now acceptable for publication, you may indicate that here to bypass the “Comments to the Author” section, enter your conflict of interest statement in the “Confidential to Editor” section, and submit your "Accept" recommendation.

Reviewer #2: All comments have been addressed

2. Is the manuscript technically sound, and do the data support the conclusions?

Reviewer #2: Yes

3. Has the statistical analysis been performed appropriately and rigorously? 

Reviewer #2: Yes

4. Have the authors made all data underlying the findings in their manuscript fully available?

Reviewer #2: Yes

5. Is the manuscript presented in an intelligible fashion and written in standard English?

Reviewer #2: Yes

6. Review Comments to the Author

Reviewer #2: The authors have addressed my concerns. The patient population of osteoarthritis patients is somewhat atypical, but I am willing to accept that. The results with respect to the low levels of TGF-beta are very interesting.

7. PLOS authors have the option to publish the peer review history of their article (what does this mean?). If published, this will include your full peer review and any attached files.

Reviewer #2: **Yes: **ManfredHarth

---

## [Editor Report · Acceptance letter]

22 Dec 2021

PONE-D-20-38558R2 

TGF-β1 in plasma and cerebrospinal fluid can be used as a biological indicator of chronic pain in patients with osteoarthritis 

Dear Dr. Chen:

I'm pleased to inform you that your manuscript has been deemed suitable for publication in PLOS ONE. Congratulations! Your manuscript is now with our production department. 

Kind regards, 

on behalf of

Dr. Firas H Kobeissy 

Academic Editor

PLOS ONE